# Vanishing Privacy: Fast Gradient Leakage Threat to Federated Learning

## Abstract

In the federated learning (FL) framework, clients participate in collaborative learning tasks under the coordination of a central server. Clients train local submodels using their own data and share gradients with the server, which aggregates the gradients to achieve privacy protection. However, recent research has revealed that gradient inversion attacks (GIAs) can leak private data from the shared gradients. Prior work has only demonstrated the feasibility of recovering input data from gradients under highly restrictive conditions, such as when dealing with high-resolution face datasets, where GIAs often struggle to initiate attacks effectively, and on object datasets like Imagenet, where they encounter limitations, primarily manifested in their ability to handle only small batch sizes and high time costs. As a result, we believe that implementing GIAs on high-resolution face datasets with large batch sizes is a challenging task. In this work, we introduce **F**ast **G**radient **L**eakage (FGL), which enables rapid image recovery across various network models on complex datasets, including the CelebA face dataset (1000 classes, 224×224 px). We also introduced StyleGAN as prior knowledge for images and achieved FGL with a batch size of 60 in experiments (constrained by experimental hardware). We further propose a joint gradient matching loss, where multiple distinct matching losses collectively contribute to clarifying the attack direction and enhancing the efficiency of the optimization process. Extensive experimentation validates the feasibility of our approach. We anticipate that our proposed method can serve as a valuable tool to advance the development of privacy defense techniques.

## 1 Introduction

Federated Learning (FL) (Li et al. (2020), McMahan et al. (2016)) aims to train high-quality global models while ensuring client privacy. In this framework, clients only use their local data (Melis et al. (2019), Shokri et al. (2017)) for training and share weights or gradients to update the global model, reducing the flow of user data and enhancing data privacy and security (Tan et al. (2022), Karimireddy et al. (2020), Chilimbi et al. (2014), Konený et al. (2016), Yang et al. (2019b)). Therefore, FL can be applied in privacy-sensitive domains such as medical data (Brisimi et al., Sadilek et al. (2020)). Hospitals can obtain a collaboratively trained global model without the need to share patient data. This approach addresses critical concerns such as data privacy, data security, data access control, and heterogeneous data access, enabling multiple participants to create a shared, powerful machine learning model without sharing data.

In most cases, the federated learning framework is considered an effective method to prevent privacy leakage. Nevertheless, recent research (Geiping et al. (2020), Yin et al. (2021), Zhao et al. (2020) Zhu & Han (2020), Wen et al. (2022)) has shown that the shared gradients contain a significant amount of sensitive information, and attackers can exploit gradient leakage to obtain client's private data. This type of attack, known as gradient inversion attacks (GIAs), has evolved to the point where it can achieve pixel-level image reconstruction. This poses a severe threat to privacy security in federated learning. While GIAs have made some progress in FL attacks, they still face limitations. For instance, using CNN models makes it challenging to carry out attacks on high-resolution face datasets (Yin et al. (2021)). Additionally, they encounter difficulties in conducting attacks with large batch sizes, along with issues related to low attack efficiency and high time costs.

We propose a GAN-based GIAs method called **F**ast **G**radient **L**eakage (FGL) to address the challenges mentioned earlier. Our technique is inspired by the field of model inversion attacks (MIAs) (Fredrikson et al. (2015), Zhang et al. (2020), He et al. (2019)), which shares similarities with our domain, yet MIAs have a longer research history, with many techniques being ahead of Gradient Inversion Attacks (GIAs). MIAs utilize GANs as prior knowledge for images, avoiding the need to synthesize images from noise; instead, the optimization process fine-tunes existing images until the target image is reached. Moreover, MIAs significantly improve attack success rates through strategies like rotation transformations and image selection. In a white-box setting, PPA (Struppek et al.) achieves large-batch and high-accuracy attacks on face datasets, a feat challenging for GIAs. Inspired by MIAs, we introduce related techniques into GIAs.

Building upon previous research, we conceptualize GIAs attacks as an optimization problem (Yin et al. (2021)). However, due to the difficulty in optimizing high-resolution images with GIAs' gradient matching losses, our main challenge lies in overcoming the tendency for the optimization process to fall into local optima. To address this challenge, we break away from the limitations of previous studies and propose the concept of a joint gradient matching loss function. Different gradient matching losses guide the optimization process from different perspectives, allowing for simultaneous optimization from multiple angles, thereby making the optimization results more likely to approach the global optimum.

Our goal is to quickly recover private images from clients. To reduce time overhead, we adopt StyleGAN (Karras et al. (2019), Karras et al. (2020)) as prior knowledge for images, allowing initial images to be fine-tuned rather than synthesized from noise, significantly reducing time costs. By introducing our proposed joint gradient matching loss function, the attack epoch is significantly reduced, making the optimization process simpler and faster. The combination of these two factors enhances attack efficiency. Additionally, with the optimization process becoming simpler, the batch size of attacks has also increased. Compared to previous methods, our approach has achieved significant improvements in both attack time and batch size.

Compared to GI, we introduced pre-trained StyleGAN as prior knowledge for images, combined with a joint gradient matching loss. This approach allowed convergence within a very small number of epochs (70 in our experiments), significantly reducing the time overhead. More importantly, we successfully conducted attacks on high-resolution ($224 \times 224$px) face images for the first time under CNN architecture, with a batch size of up to 60 (limited by hardware).

The main contributions are summarized as follows:

- For the first time, we have employed an optimization-based approach on a CNN architecture to achieve the reconstruction of high-resolution facial datasets.
- We propose a joint gradient matching loss that combines multiple advantages and significantly improve reconstruction quality.
- We have introduced a selection strategy that, when combined with multi-seed optimization strategies, enhances the quality of reconstructed images.

## 2 RELATED WORK

### 2.1 IMAGE SYNTHESIS.

The task of synthesizing images from neural networks has been a long-standing challenge, and Generative Adversarial Networks (GANs) (Zeng & Long (2022), Radford et al. (2015), Salimans et al. (2016), Brock et al. (2018), Gulrajani et al. (2017)) have achieved remarkable success in this field. The initial GAN (Zeng & Long (2022)) often faced issues with instability and training difficulties. However, techniques proposed by (Radford et al. (2015)) have addressed the stability problems in GAN training, and the improved WGAN (Gulrajani et al. (2017)) has enhanced training stability while mitigating issues like mode collapse. Despite these improvements, WGAN still generated low-quality images, prompting the proposal of WGAN-GP (Gulrajani et al. (2017)) as a further enhancement. DCGAN (Radford et al. (2015)) introduced a more stable architecture for training GANs and demonstrated that adversarial networks can learn meaningful image representations for supervised learning tasks. StyleGAN (Karras et al. (2019)) introduced style transfer, enabling intuitive control over synthesis at various scales. Building on these advancements, StyleGAN2 (Karras

et al. (2020)) addressed several image quality issues in StyleGAN, resulting in further improvements in image synthesis.StyleGAN3 (Karras et al. (2021)) made significant progress by addressing the reliance on absolute pixel coordinates in the typical synthesis process of GANs, thus opening new possibilities for video and animation synthesis. Additionally, BigGANs (Brock et al. (2018)) achieved a breakthrough by training GANs on the complex ImageNet (Deng et al. (2009)) dataset, significantly advancing the state-of-the-art in GAN research. The significant progress in GANs has brought about a revolution in image synthesis, enabling diverse applications ranging from art generation to the production of highly realistic images. However, certain challenges persist, such as optimizing GANs for specific tasks, ensuring scalability, and effectively handling large-scale datasets. As the research in GANs continues to advance, we can anticipate even more thrilling developments in the realm of image synthesis and generation.

## 2.2 PRIVACY LEAKAGE VIA GRADIENT

Recently, the field of privacy attacks in federated learning has seen significant advancements.(Zhu & Han (2020)) proposed a gradient-based privacy attack method, allowing attackers to reconstruct users' private data by matching exchanged gradients in the federated learning scenario. This poses substantial challenges to privacy and security in federated learning (Konený et al. (2016), Wang et al. (2019), Reisizadeh et al. (2019),). Building on this work, (Zhao et al. (2020)) improved the method by introducing a label inference technique, enhancing the attack efficiency. However, both methods are only applicable to shallow networks (Lecun et al. (1998)) trained on low-resolution data (Krizhevsky (2009)). Previous GIAs commonly utilized the $L_2$ norm for gradient matching. (Geiping et al. (2020)) proposed cosine similarity for gradient matching, achieving promising results and revealing vulnerabilities even in large-scale datasets trained on non-smooth networks like ResNet-152 (He et al. (2016)).(Zhu & Blaschko (2021)) introduced a novel method that advanced the understanding of GIAs. (Yin et al. (2021)) further improved label inference with GradInversion, incorporating an image regularization term to enhance image fidelity. Their approach demonstrated success in revealing privacy images with batch size ranging from 8 to 48 on large networks trained on ImageNet. In the pursuit of stronger attacks, (Huang et al. (2021)) evaluated the work of (Yin et al. (2021)) and pointed out two strong assumptions (BatchNorm statistics and private labels), suggesting that relaxing these assumptions significantly reduces the attack capability.(Hatamizadeh et al.) further demonstrated the feasibility of GIAs on vision transformers (ViTs). More recently, (Li et al.) proposed a GAN-based gradient inversion attack, capable of revealing privacy images under various gradient defenses while maintaining good image quality.

## 3 METHODOLOGY

In this section, we provide a detailed introduction to the FGL method. We first establish a threat model in Section 3.1. Then, we explain the definition of our objective function and the optimization methods employed in Section 3.2. Finally, we present a comprehensive overview of the innovative components in our approach in Section 3.3. The overall architecture is illustrated in Figure 1.

### 3.1 THREAT MODEL

In both federated learning algorithms, FedSGD and FedAvg, we assume that the attacker functions as an honest but curious server. The attacker is endowed with the capability to receive model weights $w$ and gradients $\triangle W$ transmitted by the clients. The adversary's goal is to deduce sensitive information from the client's private data by scrutinizing these parameters. It is crucial to emphasize that the server is prohibited from unilaterally modifying the initial model sent to the client (Fowl et al., Boenisch et al.). Additionally, the adversary may leverage publicly available resources such as common datasets and openly accessible pre-trained models, but their computational resources are limited.

### 3.2 OBJECTIVE FUNCTION

Considering a network with weight parameters $W$ and a gradient update $\triangle W$ obtained from a batch of ground truth images $x^*$ and their corresponding labels $y^*$, our optimization algorithm (Eqn.1 -

Figure 1: Overview of our proposed FGL method. Our proposed FGL method comprises three main stages. In the sampling stage, we adopt a multi-seed optimization approach, where we simultaneously perform optimization with multiple random seeds. During the optimization stage, we leverage a novel joint gradient matching loss function and gradient normalization technique.In the final selection stage, we carefully choose the most representative and successful results from the optimized set.

Eqn.4) aims to find an optimal solution.

$$\hat{x} = T_{trans}(S_{init}(G(z))) \tag{1}$$

$$\hat{z}^*_{seed} = arg\min_z M_{grad}(\hat{x}; N_{grad}(\Delta W, \Delta W')) \tag{2}$$

$$\hat{z}^*_{best} = S_{final}(\hat{z}^*_{seed_1}, \hat{z}^*_{seed_2}, \hat{z}^*_{seed_3} \dots \hat{z}^*_{seed_n}) \tag{3}$$

$$\hat{x}^*_{best} = G(\hat{z}^*_{best}) \tag{4}$$

In this context, we provide the following terminological explanations: $z$ represents the latent space, $G(\cdot)$ denotes the GAN, $S_{init}(\cdot)$ signifies the initial point selection strategy, $T_{trans}(\cdot)$ stands for increasing robustness by transformations, $N_{grad}(\cdot)$ indicates the gradient normalization operation, and $M_{grad}(\cdot)$ is our proposed new gradient matching loss. $\hat{z}^*_{seed}(\cdot)$ refers to the initial seed result, while $S_{final}(\cdot)$ represents the strategy for selecting the optimal result $\hat{z}^*_{best}$. Ultimately, we use the synthesized $\hat{x}^*_{best}$ as the outcome of the attack.

## 3.3 FAST GRADIENT LEAKAGE

In this section, we will systematically present our contributions and technical details, following the process of the GIAs.

**Selection Strategy**. The selection of the initial point plays a crucial role in GIAs. To exploit the potential of the initial point, we propose four selection strategies, which consist of both initial point selection and corresponding representative result selection strategies that need to be applied together.

The first strategy involves selecting points with high confidence scores $F(x)$ according to the target model as the initial points and final points. The second strategy entails choosing points with a small $L_2$ distance between the pseudo-gradient $\triangle W'$ and the true gradient $\triangle W$ as the initial points and final points. The third strategy entails selecting points with a cosine similarity close to one between the pseudo-gradient $\triangle W'$ and the true gradient $\triangle W$ as the initial points and final points. The fourth strategy involves selecting points with both a small $\mathcal{L}_2$ distance between the pseudo-gradient $\triangle W'$ and the true gradient $\triangle W$ and a cosine similarity close to one as the initial points and final points.

Among these strategies, the first initial point strategy yields the best results in our work. It is worth noting that the first strategy requires the use of a model that has already converged, while the other three strategies are applicable to all models.

**label inference.** After selecting highly attack-oriented initial points, the next step is the label inference (Qu et al. (2019), Fu et al.), where successfully obtaining the true labels greatly enhances the success rate of the attack. Based on the analysis of (Zhao et al. (2020)), under the premise of using cross-entropy as the loss function, it is possible to infer the true labels using shared gradients.

Assuming there are n classes in the dataset, denoting the $i$-th output of the model as $z_i$ , the loss as $\partial L\left(F\left(x^*\right), y^*\right)$. The label inference function can be expressed as follows:

$$\hat{y} = argsort\left(\bigtriangledown_z \mathcal{L}\left(F\left(x^*\right), y^*\right)[:1]\right) \tag{5}$$

The above analysis is based on the scenario where batch size one. However, when batch size is greater than one, there is information loss due to gradient summation. Using the label inference based on the batch size one leads to a higher error rate. To address this issue, (Yin et al. (2021)) proposed batch label restoration. Based on the observation that $|V_c| \gg |V_{i \neq c}|$ (the absolute value of the negative gradient term for class $i = c$ is larger than the absolute value of the positive gradient terms for $i \neq c$), the label inference function can be written as follows:

$$\hat{y} = argsort\left(\min_m \bigtriangledown_{W_{n,i}^{FC}} \mathcal{L}(F\left(x^*\right), y^*)\right)[:K] \tag{6}$$

In batch label restoration, we identify the rows with the smallest values in the fully connected layer and sort them. The top-k rows correspond to the restored labels.

**Gradient Matching Loss.** Even if the true label $\hat{y}^*$ is inferred, attacking high-resolution facial images remains highly challenging. Previous research works have predominantly used loss functions such as $\mathcal{L}_2$ norm (Zhu & Han (2020), Zhao et al. (2020), Yin et al. (2021)) and Cosine Distance (Geiping et al. (2020)). In our work, we propose a novel loss function design strategy comprising two parts. Firstly, we define a loss function between the target function output $Y'$ and the inferred label $Y$. Secondly, we introduce the gradient matching loss between $\triangle W'$ and $\triangle W$ . We select the Poincaré distance (Struppek et al.) as the loss function for the first part. The Poincaré loss function is used to measure the distance between two tensors $y$ and $y^*$, and it is defined as follows:

$$\begin{aligned}
\mathcal{L}_{Poincare} &= d\left(y, y^*\right) \\
&= arcosh\left(1 + \frac{2||y - y^*||^2}{(1 - ||y||_2^2)(1 - ||y^*||_2^2)}\right)
\end{aligned} \tag{7}$$

In the second part, we propose a joint loss function given by $M_{grad} = \alpha_1 \mathcal{L}_2 + \alpha_2 Cosine + \alpha_3 \mathcal{L}_1$ , where $\alpha_1$ , $\alpha_2$, and $\alpha_3$ are hyperparameters.The expressions for $\mathcal{L}_2$, $Cosine$, and $\mathcal{L}_1$ are shown :$\mathcal{L}_2(\triangle W, \triangle W') = ||\triangle W - \triangle W'||_2^2$, $Cosine(\triangle W, \triangle W') = 1 - \frac{<\triangle W, \triangle W'>}{||\triangle W||_2 \cdot ||\triangle W'||_2}$, $\mathcal{L}_1(\triangle W, \triangle W') = ||\triangle W - \triangle W'||$.

**Gradient Normalization.** In some scenarios, the values of the true gradient $\triangle W$ can be extremely small, making it difficult for the pseudo gradient $\triangle W'$ to approximate the true gradient $\triangle W$ accurately. Consequently, the loss value struggles to converge, leading to suboptimal attack performance. To address this issue and enhance the effectiveness of our approach, we apply gradient normalization (Xu et al. (2019), Xiong et al. (2020), Yang et al. (2019a)). By normalizing both the true gradient $\triangle W$ and the pseudo gradient $\triangle W'$ to the same scale, we can accelerate the convergence of the loss function and enhance the attack capability of our method as in Figure 2.

**Increasing Robustness by Transformations.** To enhance the robustness of our algorithm, we introduce image transformations (Hu et al., Athalye et al. (2017), Struppek et al.) to stabilize the attack process. We define $t$ as a single transformation operation, which can include rotations, translations, cropping, resizing, and more. $T$ represents a combination of multiple $t$ operations, given by $T_{trans}(x) = t_1(x) \cdot t_2(x) \cdots t_n(x)$ . Instead of using the original image $x$ directly, we employ the transformed image $x' = T_{trans}(x)$ during the attack.

**Multi-Seed Optimization Strategies.** Although we have applied the Selection Strategy, it is inevitable that some initial points may get trapped in local optima.

To enhance the robustness of our attack method, we propose a multi-seed optimization strategy. We sample initial point combinations using different Seeds ($Seed_1$ ,$Seed_2$, $Seed_3$, ..., $Seed_n$), where each set of initial points exhibits a relatively consistent optimization direction. By employing a multi-seed optimization strategy, we have expanded the attack exploration range of FGL, consequently enhancing its attack stability.

Figure 2: Gradient regularization for resetting gradient space.

## 4 EXPERIMENTS

### 4.1 EXPERIMENTAL SETUP

In all experiments (Apart from experimenting with different datasets and model architectures.), we employed the StyleGAN2 (Karras et al. (2020)) model pretrained on the FFHQ dataset to attack the target model ResNet-18 trained on the CelebA dataset, thus simulating the setting of federated learning. Additional details and attack parameters about the experimental can be found in the Appx. 5.

To demonstrate the effectiveness of our proposed approach, we conducted ablation experiments as detailed in Section 4.2. Subsequently, we compared our FGL method with some state-of-the-art approachesin GIAs to highlight its advantages in 4.3 . Additionally, we performed experiments on large-batch GIAs in 4.4 and time cost to validate the effectiveness of our method in 4.5.

**Evaluation Metrics.** Diverging from earlier GIAs, FGL's aim is not to replicate private images, but rather to synthesize images with akin features. Consequently, conventional metrics like SSIM, MSE, and PSNR, commonly used to determine if two images are identical, find limited applicability in our attack. In order to precisely assess the faithful representation of privacy image features in synthesized images, we introduce three corresponding evaluation metrics. (i) The Top-1 and Top-5 accuracy rates computed by Inception-v3. (ii) The feature distance $D_{inc}$ between synthesized and real images computed by Inception-v3. (iii) The feature distance $D_{face}$ between synthesized and real images computed by FaceNet.

### 4.2 ABLATION STUDIES

The purpose of the ablation experiments is to analyze the roles of different components in our proposed method. We progressively incorporate our proposed method into the optimization objective function and conduct quantitative analysis of the data in Table 1 as well as qualitative analysis based on visual observations Figure 3. We conducted a more detailed ablation study on the joint gradient matching function in Appx. 5.

| $L_{grad}(\hat{x}; \triangle W, \triangle W')$ | Image Reconstruction Metric | | | |
|---|---|---|---|---|
| | TOP-1 ↑ | TOP-5↑ | Dinc↓ | Dface↓ |
| $L_2$ | 0.0 | 0.0 | 1.0 | 1.52 |
| $+S_{init}$ | 0.0 | 0.0 | 0.92 | 1.20 |
| $+S_{final}$ | 0.12 | 0.2 | 0.82 | 1.16 |
| $+T_{trans}$ | 0.16 | 0.28 | 0.88 | 0.92 |
| $+M_{grad}$ | 0.60 | 0.80 | 0.66 | 0.82 |
| $+N_{grad}$ | 0.72 | 0.76 | 0.74 | 0.72 |
| $+M_{seed}$ | **0.88** | **0.96** | **0.72** | **0.72** |

Table 1: a quantitative comparison of different components of FGL.

**Adding** $S_{init}$**.** Choosing multiple initial points enhances the robustness of FGL, resulting in a more stable attack. This approach mitigates the impact of random initial points and improves the success rate of the attacks.

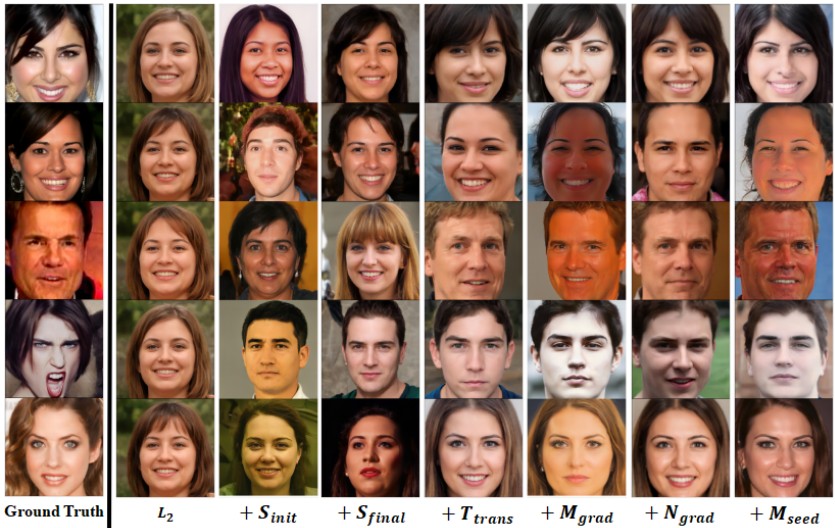

Figure 3: In the ablation study, we qualitatively compared the effects of adding each proposed loss to the optimization objective function.

**Adding $S_{final}$.** By introducing $S_{final}$, we can select the best-performing result from multiple initial points as the final output.

**Adding $T_{trans}$.** Adding transformation operations to images can enhance their robustness. Among these operations, image cropping plays a major role by removing background interference and enhancing the recognition accuracy of the target model.

**Adding $M_{grad}$.** We have improved the attack effectiveness by replacing the previous $\mathcal{L}_2$-only approach with a novel gradient matching function. By employing a joint loss function, such as $\alpha_1 \mathcal{L}_2 + \alpha_2 Cosine + \alpha_3 \mathcal{L}_1$ as the gradient matching function, the optimization process aims to minimize not only the $\mathcal{L}_2$ distance but also maximize the cosine similarity close to one, while minimizing the $\mathcal{L}_1$ distance. These combined optimization angles enable the points to approach global optima instead of being trapped in local optima.

**Adding $N_{grad}$.** Being able to attack images that are inherently difficult to attack, thereby increasing the overall attack success rate.

**Adding $M_{seed}$.** Taking into account the varying optimization difficulties among different seeds, the use of a multi-seed strategy provides an alternative perspective for enhancing the robustness of FGL.

**Different network architectures.** We investigate the impact of different network architectures on our method by conducting batch size 5 GIAs on ResNet-18 (He et al. (2016)), ResNet-152 (He et al. (2016)), and DenseNet-169 (Huang et al. (2017)), as shown in Figure 4. We observed that the shallowest model, ResNet-18, performed the best, followed by ResNet-152 with 152 layers, and the worst performance was exhibited by DensNet-169.

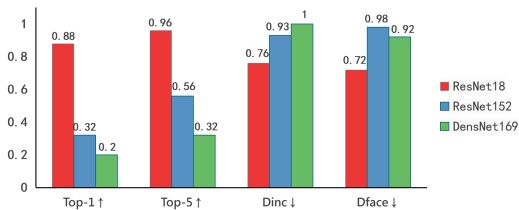

Figure 4: We conducted attacks on different network architectures and observed their visual performance.

## 4.3 COMPARISON WITH THE STATE-OF-THE-ART

| Methon | Image Reconstruction Metric | | | |
|---|---|---|---|---|
| | Top-1↑ | Top-5↑ | $D_{inc}$ ↓ | $D_{face}$ ↓ |
| DLG | 0.0 | 0.0 | 1.00 | 1.82 |
| GI | 0.0 | 0.0 | 0.31 | 1.68 |
| Fishing | 0.0 | 0.0 | 0.36 | 1.63 |
| GIAS | 0.0 | 0.0 | 0.32 | 1.62 |
| **FGL(Ours)** | **1.0** | **1.0** | **0.21** | **0.83** |

Table 2: GIAs on CelebA Dataset: A Comparative Study with State-of-the-Art Methods.

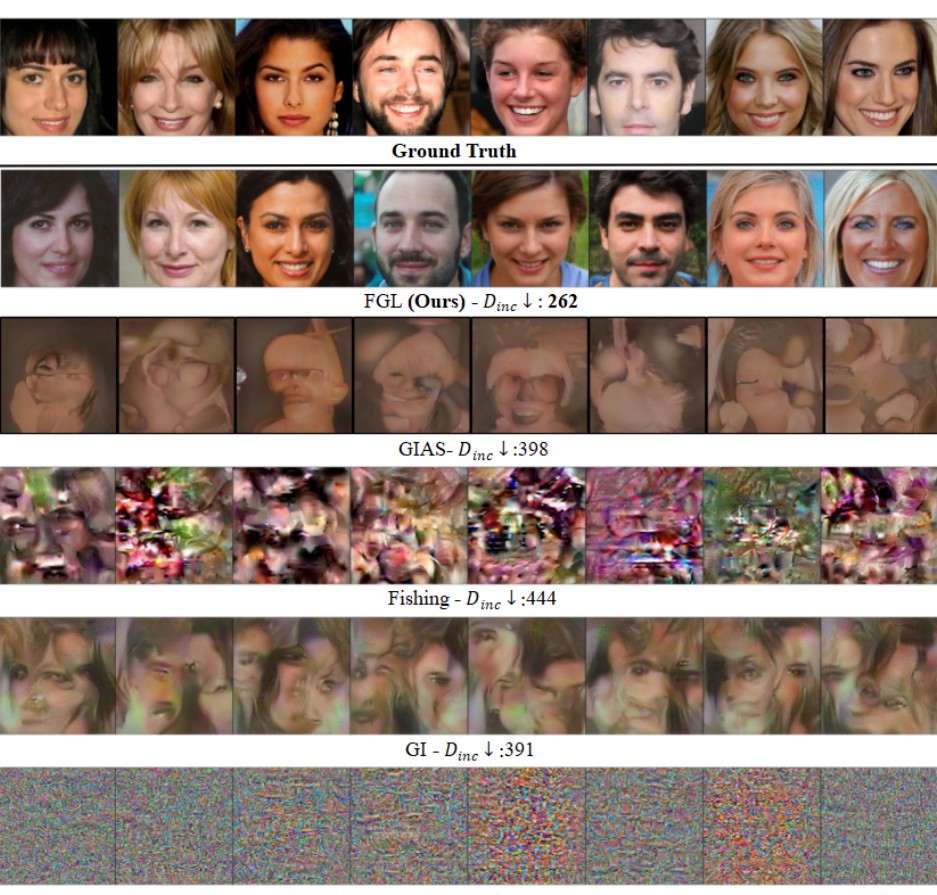

Figure 5: CelebA batch gradient inversion for ResNet-18 visual comparison with state-of-the-art methods.Our method outperforms others in terms of overall image quality and capturing fine details.

To ensure the optimal performance of the baseline method, we conducted attacks with a batch size of one. We summarize both qualitative (Figure 4) and quantitative results (Table 2). We also validated the performance of FGL on ResNet-152 and DenseNet-169 in Appx. 5. Additionally, to verify the effectiveness of FGL under various data distributions, we selected two extreme cases with entirely different data distributions for attack experiments in Appx. 5 .

**Attack Baselines.** We compare our method against four existing approaches: (i) Deep Leakage from Gradients (DLG) (Zhu & Han (2020)), (ii) GradInversion (GI) (Yin et al. (2021)), (iii) Fishing for User Data (Fishing) (Wen et al. (2022)) and (iiii) GIAS (Jeon et al.). GIAS (iiii) constitutes a generative method, serving as a closely aligned baseline model to FGL. In line with FGL's attack configuration, we also employed StyleGAN, trained on the FFHQ dataset, to generate images suitable for the CelebA dataset, thereby simulating distribution shift scenarios. To ensure a fair

comparison, I employed a ResNet-18 model with an accuracy of 86.38% on the CelebA dataset as the target model for each approach. I conducted an equivalent number of runs for the remaining methods, following the same selection strategy as FGL, utilizing different random seeds each time. The best-performing result among these runs was ultimately chosen as the conclusive outcome.

Through qualitative and quantitative comparisons, we can clearly observe FGL outperforms prior art both visually (Figure 4)) and numerically (Table 2) on the facial dataset. Previous methods rarely focused on attacking high-resolution facial images, and when we attempted to apply these methods to facial datasets, achieving remarkable attack performance was challenging. Among the four comparative methods, only the GI (Yin et al. (2021)) and Fishing (Wen et al. (2022)) method managed to capture the outline of the images, providing a rough representation of facial features, albeit with incorrect positions and lacking details. The synthesized images from the other methods DLG (Zhu & Han (2020)) only consisted of indistinguishable pixels. Compared to similar methods (Jeon et al.) that did not demonstrate effective attacks when faced with distributional shifts, our proposed novel unified matching loss makes the optimization direction more explicit. This approach proves to be more adept at avoiding local optima. Additionally, employing multiple seed optimization strategies enables us to transcend the limitations of single-seed optimization. By incorporating a selection strategy, we can identify results that are more representative.

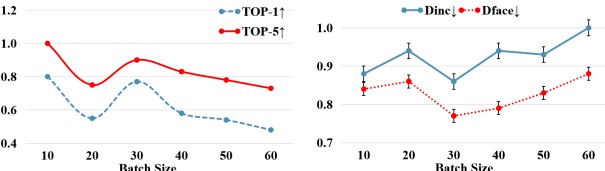

Figure 6: The impact of changing the batch size on different evaluation metrics. For the convenience of observing the variation trend of $D_{inc}$, we normalized it.

## 4.4 EFFECT OF SCALING UP THE BATCH SIZE

In our method, we conducted attacks with a batch size 60 on randomly selected CelebA images. The data results are shown in Figure 6 and Figure 13, illustrating the effectiveness of our approach.

It can be observed that at batch size 20, some images prove challenging to attack, resulting in a decline in performance. However, as we increase the batch size to 30, the performance stabilizes and rebounds. Even with a batch size 60 attack, our method achieves a Top-1 accuracy of 0.483, demonstrating its consistent and strong performance.

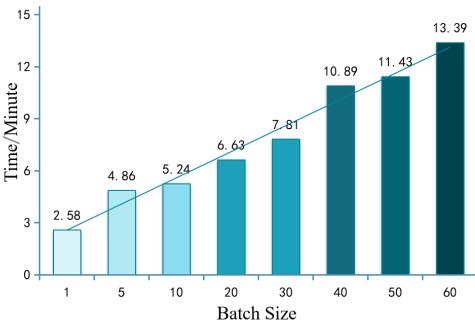

Figure 7: Time cost of our method for GIAs at different batch size.

## 4.5 TEMPORAL COST ANALYSIS

To evaluate the time cost, we conducted detailed statistics, as shown in Figure 7. DLG struggled to attack the CelebA dataset, so we used the Cifar-10 dataset where they performed well as an example. For DLG, attacking with a batch size of 4 required 1173 iterations and took approximately 17.56

minutes. On the other hand, GI required 10000 iterations and 23.99 minutes to attack the CelebA dataset with a batch size one, while Fishing required 24000 iterations and 23.17 minutes for the same setting. GIAS requires 4800 iterations to complete one attack, taking 24 minutes. However, for a batch size one attack under the same conditions, our method only took 2.58 minutes to complete.

Importantly, even when using a large batch size 60 for the attack, the required time cost did not exponentially increase but only amounted to 13.99 minutes. Our method consistently exhibited lower time costs at each stage compared to the other methods, emphasizing the efficiency advantage of our approach in attacks.

## 5 CONCLUSIONS

In this work, we proposed FGL, which allows for the rapid disclosure of a large amount of privacy information across different data distributions. These experimental findings underscore the significant potential of GAN-based GIAs in the field of privacy protection. We believe that our contributions will stimulate advancements in the field of privacy-preserving deep learning and contribute to the construction of more secure and privacy-aware deep learning systems.

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
