## APPENDIX A - ADDITIONAL DETAILS AND ANALYSIS

### HARD- AND SOFTWARE DETAILS

This experiment was conducted on a server equipped with 4 Tesla P100 GPUs and 256GB of memory. All models in the experiment were implemented using PyTorch. The training process further relied on CUDA 11.4, Python 3.8.10, and PyTorch 1.12.1 with Torchvision 0.13.1.

### IMPLEMENTATION DETAILS

We employ StyleGAN2 (Karras et al. (2020)) with pre-trained weights from the FFHQ256 dataset for style transfer. We trained three target models, ResNet-18 (He et al. (2016)), ResNet-152 (He et al. (2016)), and DenseNet-169 Huang et al. (2017), on the CelebA dataset. They achieved accuracies of 86.38%, 87.35%, and 85.39% respectively on the test set. The selected initial points $z$ follow a Gaussian distribution with a dimensionality of 512. We evaluate the attack using Inception-v3 (Szegedy et al. (2016) )and FaceNet (Schroff et al. (2015)) as evaluation models. We utilize Adam optimizer (Kingma & Ba (2014)) with an initial learning rate of 0.001 and $\beta$=(0.9, 0.999). The joint loss function is defined as $M_{gard} = \alpha_1 \mathcal{L}_2 + \alpha_2 Cosine + \alpha_3 \mathcal{L}_1$, where $\alpha_1$=0.5, $\alpha_2$=0.5, and $\alpha_3$=0.5.

| Real Lable | Label Restoration | |
| --- | --- | --- |
| | Inference results | Label Restoration Accuracy |
| [569] | [569] | 100.0% |
| [25,759] | [25,759] | 100.0% |
| [405,556,587] | [405,556,587] | 100.0% |
| [405,566,587,532] | [405,566,587,914] | 75.0% |
| [768,996,199,28] | [768,199,224,87] | 50.0% |
| [768,996,199,28,367] | [768,199,367,224,87] | 60.0% |
| [768,996,199,28,367,390] | [768,199,367,390,224,87] | 66.7% |
| [768,996,199,28,367,390,783] | [768,199,367,390,224,87,539] | 57.1% |
| [768,996,199,28,367,390,783,765] | [768,199,367,390,765,224,87,539] | 62.5% |

Table 3: Accuracy in inferring multiple labels from the CelebA dataset.

### ATTACK PARAMETERS

In this section, we provide a detailed description of the parameter settings used in FGL experiments. To present the information more clearly, I have organized it into a table, as shown in Table 4.

| Experiment Name | Target Id | Batch Size | Multi-Seed | Init-Point | Epoch | Vgrad Parameter |
| --- | --- | --- | --- | --- | --- | --- |
| Ablation experiment | [354, 788, 280, 556, 568] | 5 | 5 | 3000⟶4 | 70 | 1 |
| Joint loss function | random[30] | 30 | 6 | 5000⟶6 | 70 | 1 |
| Different network architectures | [354, 788, 280, 556, 568] | 5 | 5 | 3000⟶4 | 70 | 1 |
| Gradient Regularization | [556, 28, 379, 672, 81, 652, 718,848] | 8 | 2 | 3000⟶2 | 200 | $10^{-5},10^{-4},10^{-3},10^{-2},0.1,1$ |
| Comparison with the state-of-the-art | [556, 28, 379, 672, 81, 652, 718,848] | 1 | 2 | 3000⟶2 | 70 | 1 |
| Different Batch Size | random[10,20,30,40,50,60] | 10,20,30,40,50,60 | 6 | 5000⟶6 | 70 | 1 |

Table 4: In the experiment, we configured various parameters.

### LABEL INFERENCE

In our experiments, we observed a phenomenon different from previous research: label inference poses unique challenges in the context of GIAs. Firstly, (Zhao et al. (2020)) is the first work to introduce label inference in GIAs, demonstrating high accuracy in inferring labels for individual samples. However, this approach is not suitable for large batches of data. To address this issue, (Yin et al. (2021)) proposed a label inference technique that is effective for large batches and validated its performance on the ImageNet dataset.

However, our experimental focus was on the CelebA dataset, consisting of $224 \times 224$-pixel images of faces. We directly applied the batch label inference method proposed by (Yin et al. (2021)), but found it unsuitable for CelebA. While it performed well with a batch size 3 or smaller, achieving 100% inference accuracy, its performance degraded when the batch size exceeded 3, leading to a decrease in inference accuracy. We also investigated the inference performance for different labels,

as shown in Table 3. This had implications for our experiments. Consequently, when evaluating the effectiveness of our method, we opted to use ground truth labels directly rather than employing the label inference method.

## APPENDIX B - ADDITIONAL EXPERIMENTS

### RELATIONSHIP BETWEEN GRADIENT REGULARIZATION AND IMAGE QUALITY.

During the attack on the CelebA dataset, we encountered the issue of generated image gradients being too small, which made gradient matching challenging. To address this problem, we introduced the technique of gradient regularization in our research, which enables better optimization by processing gradients. However, we also observed that gradient regularization has an impact on image quality. We conducted a preliminary quantitative analysis of the gradient normalization parameter $V_{grad}$, and the results are presented in Table 5. Additionally, we showcase the image performance in Figure 8.

| $V_{grad}$ | Image Reconstruction Metric | | | |
| --- | --- | --- | --- | --- |
| | Top-1↑ | Top-5↑ | $D_{inc} \downarrow$ | $D_{face} \downarrow$ |
| $10^{-5}$ | 1.00 | 1.00 | 1.00 | 0.71 |
| $10^{-4}$ | 1.00 | 1.00 | 0.66 | 0.72 |
| $10^{-3}$ | 0.00 | 1.00 | 0.86 | 0.86 |
| 0.01 | 1.00 | 1.00 | **0.41** | 0.86 |
| 0.1 | 1.00 | 1.00 | 0.58 | 0.71 |
| 1 | **1.00** | **1.00** | 0.46 | **0.62** |

Table 5: The Impact of Different $V_{grad}$ Parameters on Image Quality.

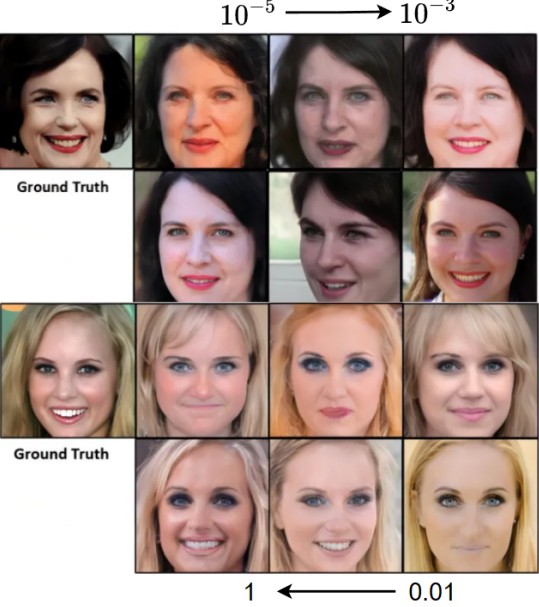

Figure 8: The Impact of Different $N_{grad}$ Parameters on Visual Image Quality.

In the experiment of gradient regularization technique, we tried different values of the regularization parameter $V_{grad}$ ranging from $10^{-5}$ to 1. Surprisingly, setting higher values for the regularization parameter did not negatively impact the image quality. On the contrary, higher parameter values were found to be more favorable for the optimization process, leading to improved image quality and higher similarity with real images. This is because both the gradient $\triangle W$ and the pseudo-gradient $\triangle W'$ undergo normalization, which preserved the matching precision without significant

degradation. Consequently, in our experiments, we selected a regularization parameter $V_{grad} = 1$ for gradient normalization.

**$M_{grad}$** JOINT LOSS FUNCTION.

In the gradient matching task, Although we have demonstrated the effectiveness of the joint gradient function through ablation experiments, we are eager to gain a deeper understanding of its underlying mechanisms. We plan to further investigate how each individual regularization term in the joint gradient function contributes to the attack, whether a specific regularization term plays a crucial role independently, or if multiple regularization terms collaboratively generate the attack effect. Additionally, we intend to explore which part of the regularization terms significantly influences the attack results. we investigated the individual contributions of the components in the joint loss function $M_{grad} = \alpha_1 \mathcal{L}_2 + \alpha_2 Cosine + \alpha_3 \mathcal{L}_1$, as well as their combined effects. The results are shown in Table 6 and Figure 9.

| $M_{grad}$ | Image Reconstruction Metric | | | |
|---|---|---|---|---|
| | Top-1↑ | Top-5↑ | $D_{inc} \downarrow$ | $D_{face} \downarrow$ |
| $\mathcal{L}_2$ | 0.23 | 0.43 | 1.00 | 1.06 |
| Cosine | 0.40 | 0.70 | 0.92 | 0.98 |
| $\mathcal{L}_1$ | 0.30 | 0.56 | **0.84** | 0.94 |
| $\mathcal{L}_2$+Cosine | 0.40 | 0.80 | 0.94 | 0.99 |
| $\mathcal{L}_2$+$\mathcal{L}_1$ | 0.43 | 0.56 | 0.91 | 1.03 |
| Cosine+$\mathcal{L}_1$ | 0.53 | 0.63 | 0.95 | 0.99 |
| $\mathcal{L}_2$+Cosine+$\mathcal{L}_1$ | **0.53** | **0.80** | 0.89 | **0.91** |

Table 6: Comparing the Impact of Joint Gradient Matching Loss on Image Quality.

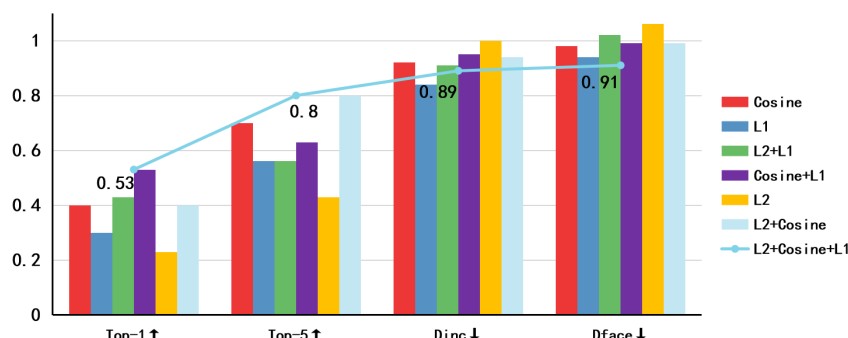

Figure 9: We conducted a study on the impact of different gradient matching loss functions on the overall attack effectiveness. In the ablation experiments, we used bar charts to illustrate the influence of different components, while for the complete joint function, we employed line charts to present the results.

In the experiments with the joint gradient loss function, we observed that each individual regularization term did not yield satisfactory results, with the cosine term slightly outperforming the others. However, when combining two regularization terms, the combination of cosine and $\mathcal{L}_1$ produced the best results. Surprisingly, the highest quality attack performance was achieved when all three regularization terms were combined simultaneously. Therefore, we believe that in the joint gradient loss function, the combined effect of multiple regularization terms plays a crucial role in achieving the optimal attack performance, rather than relying solely on individual effects or simple stacking of the terms.

FGL ON DIFFERENT DATA DISTRIBUTIONS

To validate the effectiveness of FGL in attacks under diverse data distributions, we not only conducted attacks on the Celeba dataset using the FFHQ dataset, but also performed attacks under extreme conditions using the significantly different MetFaces and AfhqDog datasets. The experimen-

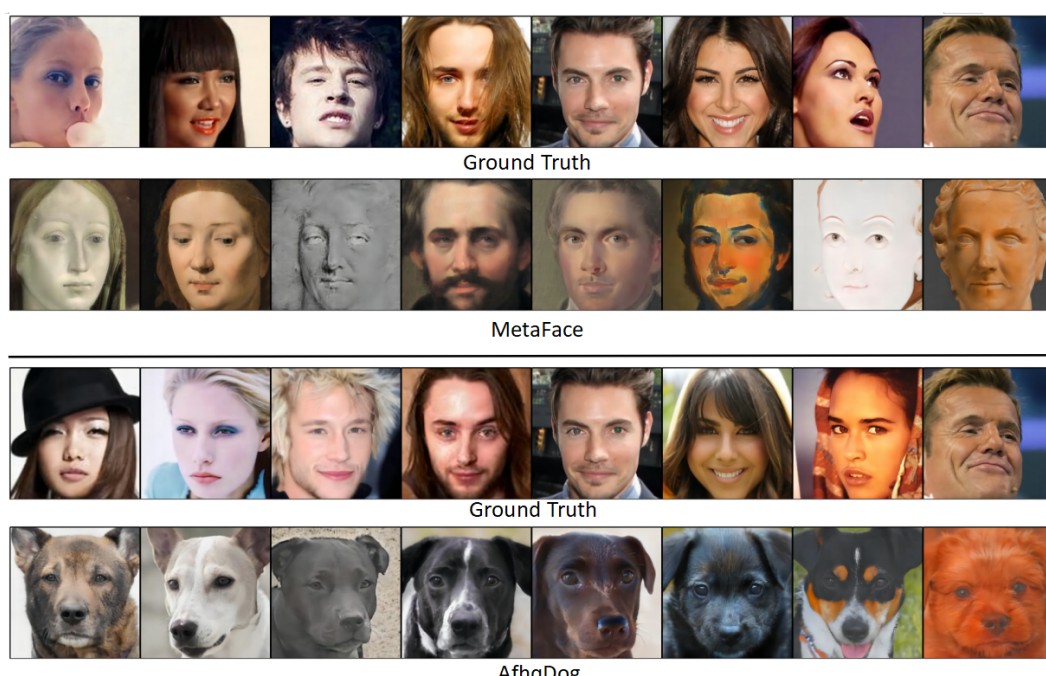

Figure 10: We conducted experiments using datasets (MetFaces and AfhqDog) with entirely different data distributions from CelebA.

tal results, as shown in Figure 10, demonstrate a noticeable resemblance between images generated from both the MetFaces and AfhqDog datasets and the real images. This affirms that FGL is capable of learning feature distributions similar to real data from datasets with distinct data distributions.

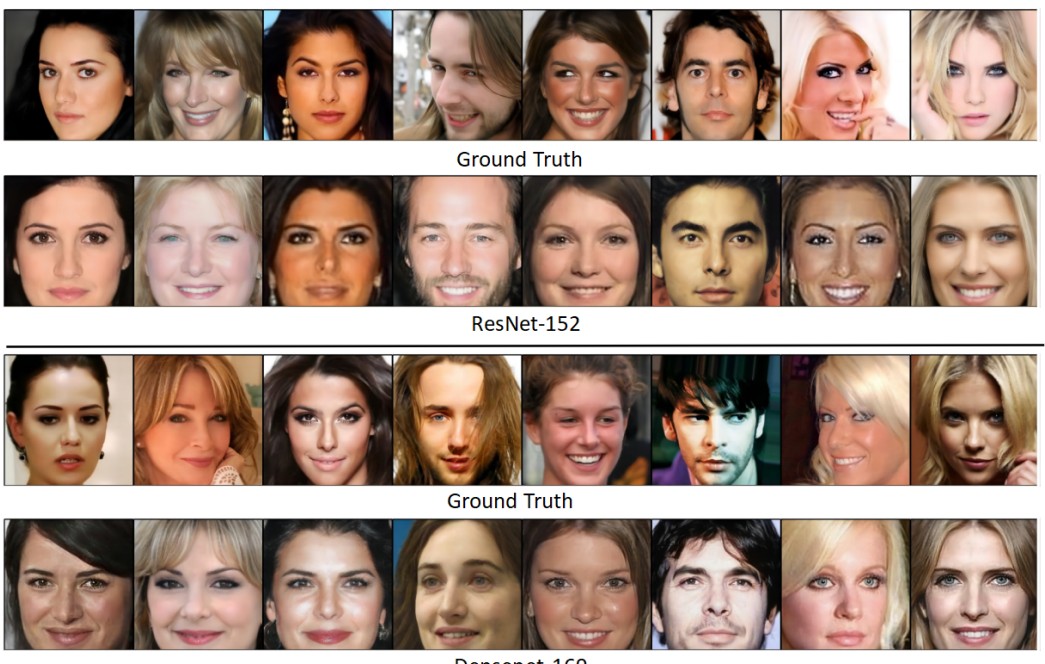

Figure 11: Experiments were conducted on deeper models, ResNet-152 and DenseNet-169.

### FGL ON DIFFERENT NETWORK ARCHITECTURES

We not only verified the effectiveness of FGL on ResNet-18, but also conducted performance tests on deeper models, ResNet-152 and DenseNet-169, to assess its practical applicability. As shown in Figure 11, the experimental results are evident: both ResNet-152 and DenseNet-169 demonstrate remarkable performance in generating images highly similar to real ones.

### FGL ON DOG DATASET

In addition to performing well on the facial dataset, FGL was also validated on an animal dataset. We trained ResNet-18 as the target model using the Stanford Dogs dataset and utilized Animal Faces-HQ Dogs (AFHQ Dogs) as the image prior to train the GAN. The experimental results are shown in Figure 14.

In our study, we observed an interesting phenomenon: humans are less sensitive to subtle differences in objects or animals, but more sensitive to variations in facial images. This heightened sensitivity towards facial images can be attributed to the frequent exposure to diverse facial representations in daily life, making us more attuned to facial changes. In contrast, for small animals like dogs, minor differences may not capture as much attention. This phenomenon is evident in the experimental examples we provide.

### FGL ON FEDAVG

Differing from FedSGD, FedAVG (McMahan et al. (2016)) performs multiple local updates before sending the model weights $w$ to the server. In our experiments, we conducted multiple trials of attacks on FedAVG using FGL. In the configuration of FedAVG, we defined some hyperparameters for local updates. Here, E (epoch) represents the local epoch, It (iteration) denotes the number of local updates, and bs (batch size) indicates the batch size for each local update. We conducted experiments with different parameters, and the results are shown in Figure 12 and Table 7 .

| E/It/bs | Distance to Original Images | | | |
|---|---|---|---|---|
| | TOP-1 | TOP-5 | $D_{inc} \downarrow$ | $D_{face} \downarrow$ |
| E=1 It=8 bs=1 | 0.500 | 0.5 | 427 | 0.94 |
| E=1 It=4 bs=2 | 0.375 | 0.625 | 398 | 1.00 |
| E=2 It=8 bs=1 | 0.25 | 0.375 | 439 | 0.97 |
| E=2 It=4 bs=2 | 0.5 | 0.5 | 449 | 1.14 |
| E=3 It=8 bs=1 | 0.375 | 0.75 | 452 | 0.98 |
| E=3 It=4 bs=2 | 0.25 | 0.25 | 502 | 1.10 |

Table 7: We conducted a series of FGL attack experiments with various parameters on FedAVG.

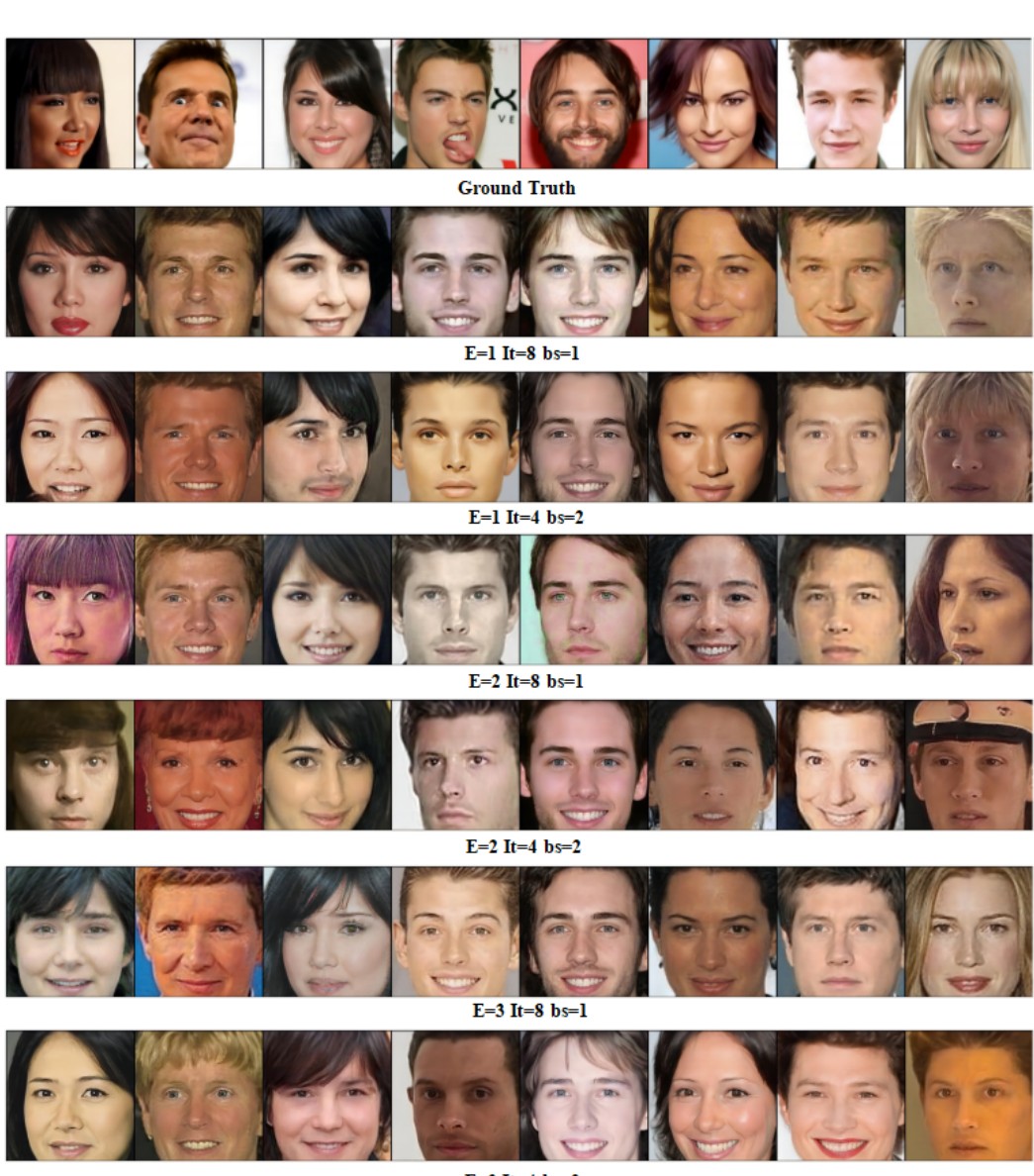

Figure 12: Here are the qualitative experimental results of FGL on FedAVG with different parameter settings.

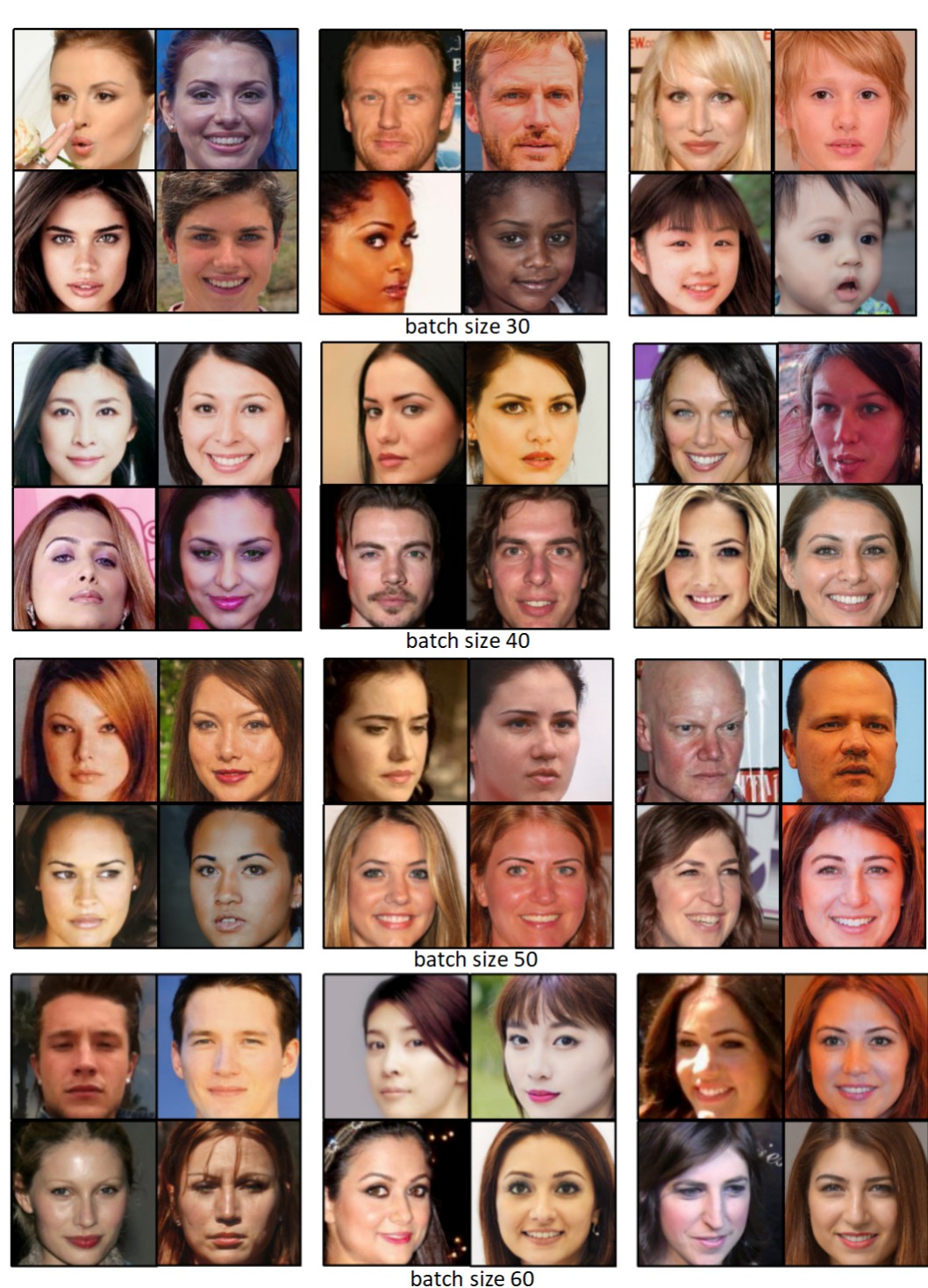

batch size 30

batch size 40

batch size 50

batch size 60

Figure 13: Provide more examples of large-batch attacks, with real images on the left and synthesized images on the right.

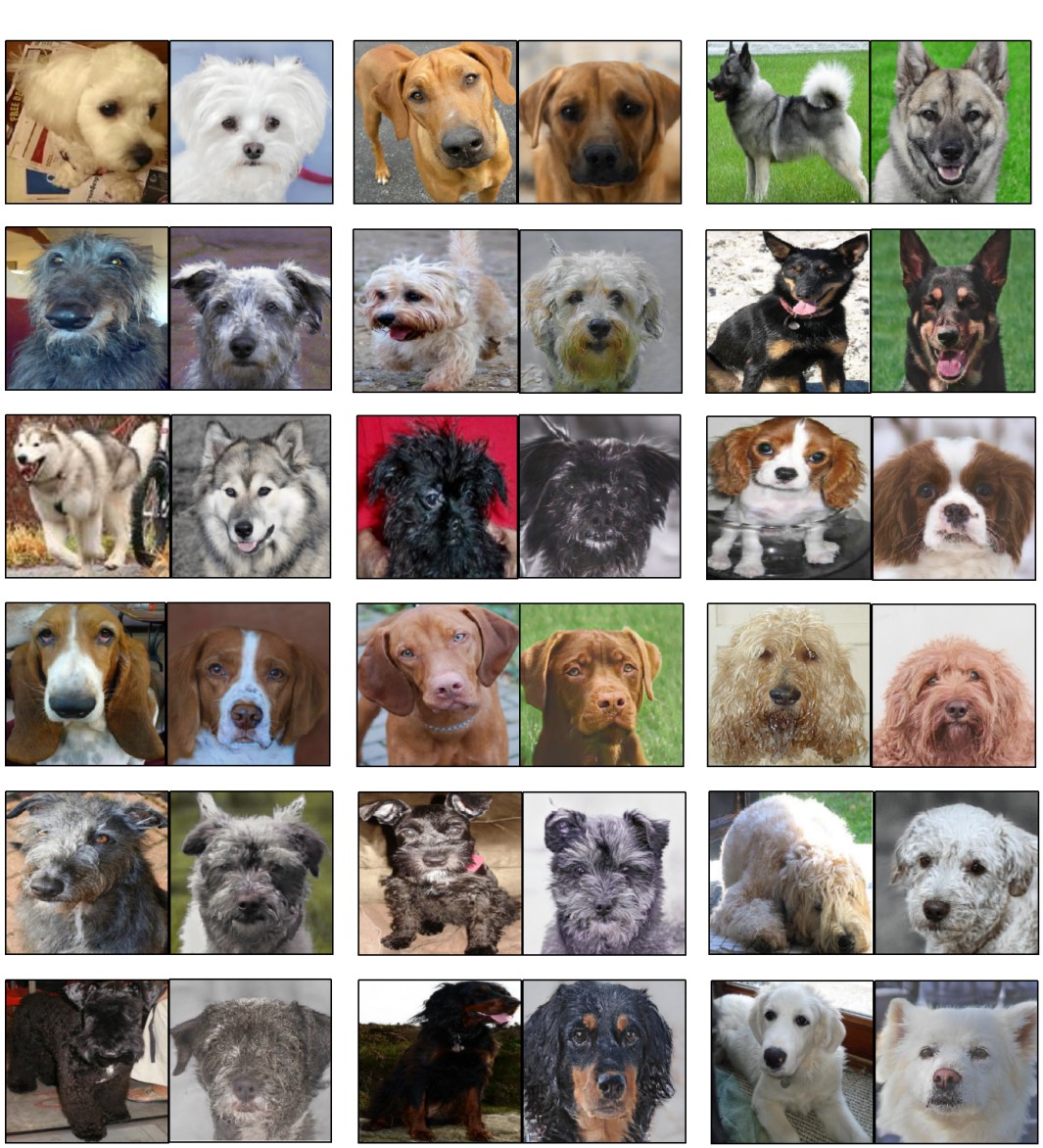

Figure 14: FGL conducted attacks on the Stanford Dogs dataset, with real images on the left and synthesized images on the right.