# OpenReview forum: "Vanishing Privacy: Fast Gradient Leakage Threat to Federated Learning"
_ICLR.cc/2025/Conference — Submitted to ICLR 2025_

### Official Review · Reviewer_WViV · 2024-10-24

**Soundness:** 1
**Presentation:** 1
**Contribution:** 1
**Rating:** 3
**Confidence:** 5

**Summary:**

This paper proposes a fast gradient inversion attack method by combining StyleGAN and multiple loss. This method has achieved, for the first time, high-resolution privacy reconstruction on the facial dataset.

**Strengths:**

1. This paper proposes a GIA method that claims to converge faster than other approaches.

2. This paper validates the effectiveness for the first time on a facial dataset.

**Weaknesses:**

This paper has significant limitations:

The experimental designs for the comparison with baselines are terrible. The authors not only changed the attack dataset but even altered the evaluation metrics. If the paper aims to demonstrate the superiority of its attack performance, it should choose the same scenario and evaluation metrics as baselines, or it is not convincing. Secondly, the reproduction of baselines is also unconvincing. To my knowledge, the performance of these baselines is not as poor as depicted in Fig. 5 of this paper. Although the paper claims to use a face dataset combined with StyleGAN, there are GIA-related studies [1-2] that also use GANs to provide prior knowledge, which the authors did not compare against.

The novelty is limited. In my view, the authors' method is mostly a combination of various existing methods, including L1 loss [3], GAN prior [1-2], and multiple seed strategies [4]. As for the Poincare loss for labels proposed in Eq. (7), I cannot understand the authors' intention because, according to the previous text, the label y has already been inferred. The paper does not introduce and analyze how to use this loss and its effect. According to the results provided in the Appendix, only about 60% of the labels are recovered correctly for 8 different labels, which is far inferior to the method proposed by Yin et al. Therefore, I cannot see the value of using this loss.

Insufficient analysis. The authors claim that less time cost is a characteristic of their method, but the presentation of this important result is only a narrative in the final section of the paper. Time comparisons should be made with certain prerequisites, such as the time taken to reach a certain quality threshold. Without any restrictions, since other baselines perform so poorly in the authors' demonstration, they could be terminated early.

Additionally, there are many other issues with the paper, including confusing formula expressions (such as the loss with a partial derivative symbol at line 217, and the 'm' in Eq. (6)); Eq. (5) uses the derivative of the model output z, but in fact, an attacker can obtain not the derivative of z (z is not a model parameter); the color scheme in Fig. 1 is terrible; and Fig. 2 lacks any detailed introduction.

[1] J. Jeon, K. Lee, S. Oh, J. Ok, et al., “Gradient inversion with generative image prior,” Proc. Adv. Neural Inf. Process. Syst., vol. 34, pp. 29898– 29908, 2021.

[2] Z. Li, J. Zhang, L. Liu, and J. Liu, “Auditing privacy defenses in federated learning via generative gradient leakage,” in Proc. IEEE Conf. Comput. Vis. Pattern Recognit., pp. 10132–10142, 2022.

[3] J. Deng, Y. Wang, J. Li, C. Shang, H. Liu, S. Rajasekaran, and C. Ding, “TAG: Gradient attack on transformer-based language models,” in Proc. Conf. Empirical Methods Natural Lang. Process., 2021.

[4] H. Yin, A. Mallya, A. Vahdat, J. M. Alvarez, J. Kautz, and P. Molchanov, “See through gradients: Image batch recovery via gradinversion,” in Proc. IEEE Conf. Comput. Vis. Pattern Recognit., pp. 16337–16346, 2021.

**Questions:**

The authors should conduct a comprehensive comparison and analysis with existing studies to validate the superiority of the proposed method. Please refer to the weaknesses section for my other concerns.

---

### Official Review · Reviewer_xmiY · 2024-10-30

**Soundness:** 2
**Presentation:** 2
**Contribution:** 2
**Rating:** 1
**Confidence:** 5

**Summary:**

This paper addresses the challenge of gradient inversion attacks (GIAs) in federated learning, which can leak private data from shared gradients. Previous work showed GIAs' limitations with high-resolution datasets and small batch sizes. The authors introduce Fast Gradient Leakage (FGL), a method for rapid image recovery on complex datasets like CelebA, using StyleGAN as prior knowledge. FGL supports larger batch sizes and employs a joint gradient matching loss to improve attack efficiency. Extensive experiments confirm FGL's feasibility, offering insights for enhancing privacy defense techniques.

**Strengths:**

1. Scalability and Efficiency: FGL demonstrates the ability to handle larger batch sizes and complex datasets, such as the CelebA face dataset, improving upon the constraints of earlier methods.

2. Integration of Prior Knowledge: By incorporating StyleGAN as prior knowledge, the method enhances the quality and speed of image recovery, showcasing an advanced technique in leveraging existing models.

3. Comprehensive Validation: The paper provides extensive experimentation to validate the feasibility and effectiveness of FGL, offering a robust foundation for future research in privacy defense techniques.

**Weaknesses:**

There are two main weaknesses in this paper.

1. The usage of GAN in the paper to construct a gradient inversion attack (GIA) framework is not innovative, as existing work has already proposed it, such as GGL [1]. The FGL in this paper is very close to that.
In addition, label inference adopts the method of Yin et al [2].; the gradient matching loss function is just a combination of L1, L2, etc., which was attempted by Deng et al. in TAG [3]; multi-seed optimization has been used in IG and GradInversion [2]. Therefore, the several innovations claimed in this paper are insignificant.

2. As shown in Figures 3 and 5 (and Appendix), the facial images recovered by FGL are very different from the original images. Without considering the background, the reconstructed portraits are very different from the original ones in terms of face shape, hair, and skin color. This is a large error at the pixel level and semantic level.
If the recovered data is used in downstream applications (such as face recognition), then the images obtained by GIAs will not have much value.

[1] Li, Zhuohang, et al. "Auditing privacy defenses in federated learning via generative gradient leakage." *Proceedings of the IEEE/CVF Conference on Computer Vision and Pattern Recognition*. 2022.

[2] Yin, Hongxu, et al. "See through gradients: Image batch recovery via gradinversion." *Proceedings of the IEEE/CVF conference on computer vision and pattern recognition*. 2021.

[3] Deng, Jieren, et al. "Tag: Gradient attack on transformer-based language models." *arXiv preprint arXiv:2103.06819* (2021).

**Questions:**

1. If the most popular diffusion model is used as prior knowledge instead of GAN, how will the results of GIAs change? Will the effect be better or worse?

2. There are so many different GAN models. Why do you choose StyleGAN instead of others in your experiments?

3. The label recovery method proposed by Yin et al. is not applicable to the case of repeated labels. How do you recover the labels when a category has multiple samples?

4. Since you use the method proposed by Yin et al. to recover the labels, what is the role of the Poincare loss function proposed in Equation (7)?

5. How should Safety, Risk, and the gradual transition between Safety and Risk at the top of Figure 1 be explained? It is not mentioned through the paper.

6. Other problems:

- Line 80: GI appears for the first time, and references should be cited;

- Line 96: The subtitle should not have a period, which is consistent with other (sub)titles;

- Line 151: The mathematical symbols of model weights $w$ and gradient $\nabla W$ used in the following text are inconsistent;

- Line 213: The title "label inference" should have its first letter capitalized;

- References in the text: Most references should be in the format of (XXX et al., 2020). Currently, brackets are nested in brackets, such as (Li et al. (2020), McMahan et al. (2016)), which makes the format of the article confusing. This may be due to confusion between the usage of `\citep{}` and `\citet{}`;

- References are missing years: The references in lines 137-138, 154-155, and 258-259 are missing years. Please check and unify the citation format of all references.

---

### Official Review · Reviewer_vuci · 2024-11-02

**Soundness:** 2
**Presentation:** 2
**Contribution:** 2
**Rating:** 3
**Confidence:** 4

**Summary:**

Gradient inversion attacks (GIA) can reconstruct private training data in local clients in federated learning, yet, prior works have limitations on handling more challenging scenarios such as high-resolution images and large batch size.
This work proposes a new gradient inversion attack method, which employs the StyleGAN to introduce prior knowledge and a gradient matching loss to help reconstruction.
Experimental results show that the proposed attack method can work on high-resolution face images with a large batch size (i.e., 60).

**Strengths:**

- The study problem is important and the motivation is clear.
- The results seem effective in reconstruction.

**Weaknesses:**

- Limited novelty and inappropriate claim. The authors claim "For the first time, we have employed an optimization-based approach on a CNN architecture to achieve the reconstruction of high-resolution facial datasets", previous works such as [1-2] already adopted GANs for introducing prior knowledge and conducted experiments on face images.
- Experimental results are not convincing. Please see Questions for details.




[1] Li, Zhuohang, et al. "Auditing privacy defenses in federated learning via generative gradient leakage." CVPR 2022.

[2] Li, Ziang, et al. "GAN you see me? enhanced data reconstruction attacks against split inference." Advances in Neural Information Processing Systems 36 (2024).

**Questions:**

- Regarding the claim "It is crucial to emphasize that the server is prohibited from unilaterally modifying the initial model sent to the client (Fowl et al., Boenisch et al.)", these two cited works actually hypothesize that the server can modify the model architecture or parameters, since the authors also assume an honest but curious server, then why the server cannot modify the initial model?
- Also, Analytic-based attack methods such as [3] present impressive reconstruction quality, which is advised to be considered for comparisons.
- Typically, we also need to evaluate the performance of attack methods against defenses like prior works. It is advised to consider some defenses [6-10] in the evaluation.
- Why use Poincaré Distance? The faces dataset used in the experiments didn’t show hierarchical relationships, can the authors please explain the rationale behind this?
- Regarding the metrics, the authors mentioned "Consequently, conventional metrics like SSIM, MSE, and PSNR, commonly used to determine if two images are identical, find limited applicability in our attack", this is not conniving as these metrics can measure the quality of reconstructions. Can the authors please explain this fruther?

[3] Liam H. Fowl, Jonas Geiping, Wojciech Czaja, Micah Goldblum, and Tom Goldstein. Robbing the fed: Directly obtaining private data in federated learning with modified models. ICLR, 2022.

[4] Mislav Balunovic, Dimitar Iliev Dimitrov, Robin Staab, and Martin T. Vechev. Bayesian framework for gradient leakage. ICLR, 2022.

[5] Yuxin Wen, Jonas Geiping, Liam Fowl, Micah Goldblum, and Tom Goldstein. Fishing for user data in large-batch federated learning via gradient magnification. ICML, 2022.

[6] Garov, Kostadin, et al. "Hiding in Plain Sight: Disguising Data Stealing Attacks in Federated Learning." ICLR 2023.

[7] Sun, Jingwei, et al. "Soteria: Provable defense against privacy leakage in federated learning from representation perspective." CVPR, 2021.

[8] Gao, Wei, et al. "Privacy-preserving collaborative learning with automatic transformation search." CVPR, 2021.

[9] Scheliga, Daniel, Patrick Mäder, and Marco Seeland. "Precode-a generic model extension to prevent deep gradient leakage." WACV, 2022.

[10] Huang, Yangsibo, et al. "Instahide: Instance-hiding schemes for private distributed learning." ICML. 2020.

---

### Official Review · Reviewer_tpuQ · 2024-11-04

**Soundness:** 2
**Presentation:** 2
**Contribution:** 2
**Rating:** 5
**Confidence:** 4

**Summary:**

The paper presents FGL, an approach that leverages pre-trained StyleGAN models and a joint gradient matching loss to enhance image reconstruction from gradient data, even with larger batch sizes and high-resolution images. It also proposes a new loss function integrating multiple gradient-matching techniques to prevent optimization from converging at local minima. FGL reduces the time required to achieve meaningful reconstruction, enabling faster and more accurate privacy leakage on complex datasets.

**Strengths:**

Using StyleGAN and a unique loss method for faster and clearer image recovery is creative and effective. The experiments part is clear and it demonstrates FGL’s advantages over other methods. This work reveals important privacy risks in federated learning, especially on sensitive data, highlighting the need for stronger defenses.

**Weaknesses:**

1) The paper mainly focuses on high-resolution facial images, which limits generalizability to other data types, such as medical or financial datasets often used in federated learning.

2) While the paper compares FGL with prior attack methods, it does not assess its effectiveness against some privacy defenses, such as gradient perturbation or differential privacy techniques. They are widely used in federated learning.

3) FGL’s batch size is constrained by experimental hardware, leaving uncertainty about how the method performs with larger groups of clients typical in federated learning. Generally, it tends to be hard to recover data on a large batch size.

**Questions:**

1) Have you considered testing FGL on other types of datasets beyond facial images, such as medical or financial data?

2) How does FGL perform against emerging privacy defenses, such as differential privacy or gradient noise injection?

3) Does FGL perform differently with architectures other than ResNet, particularly those with fewer layers or different feature extraction techniques?

4) You mention that the batch size in your experiments is limited by hardware. Is there any theoretical analysis of FGL’s performance as batch sizes increase?

---

### Meta-Review · Area_Chair_aSxo · 2024-12-21

**Metareview:**

The paper presents FGL, an approach that leverages pre-trained StyleGAN models and a joint gradient matching loss to enhance image reconstruction from gradient data, even with larger batch sizes and high-resolution images. It also proposes a new loss function integrating multiple gradient-matching techniques to prevent optimization from converging at local minima. FGL reduces the time required to achieve meaningful reconstruction, enabling faster and more accurate privacy leakage on complex datasets.

The authors did not respond to the reviewers' comments. Their reviews indicate the paper needs more work to improve.

**Additional Comments On Reviewer Discussion:**

The authors did not respond to the reviewers' comments.

---

### Decision · Program_Chairs · 2025-01-22

Reject